# Point Prevalence Survey of Antimicrobial Use during the COVID-19 Pandemic among Different Hospitals in Pakistan: Findings and Implications

**DOI:** 10.3390/antibiotics12010070

**Published:** 2022-12-30

**Authors:** Zikria Saleem, Abdul Haseeb, Brian Godman, Narjis Batool, Ummara Altaf, Umar Ahsan, Faiz Ullah Khan, Zia Ul Mustafa, Muhammad Umer Nadeem, Muhammad Junaid Farrukh, Muhammad Mugheera, Inaam Ur Rehman, Asma Fareed Khan, Hamid Saeed, Mohammad Akbar Hossain, Mohamed Raafat, Rozan Mohammad Radwan, Muhammad Shahid Iqbal

**Affiliations:** 1Department of Pharmacy Practice, Faculty of Pharmacy, Bahuddin Zakaria University, Multan 60800, Pakistan; 2Department of Clinical Pharmacy, College of Pharmacy, Umm AL-Qura University, Makkah 21955, Saudi Arabia; 3Centre of Medical and Bio-Allied Health Sciences Research, Ajman University, Ajman 346, United Arab Emirates; 4School of Pharmacy, Sefako Makgatho Health Sciences University, Ga-Rankuwa, Pretoria 0208, South Africa; 5Strathclyde Institute of Pharmacy and Biomedical Sciences, Strathclyde University, Glasgow G4 0RE, UK; 6Australian Institute of Health Innovation, Center of Health Systems and Safety Research, Faculty of Medicine, Health and Human Sciences, Macquarie University, Sydney 2109, Australia; 7Ghurki Trust Teaching Hospital, Lahore 54000, Pakistan; 8Department of Infection Prevention and Control, Alnoor Specialist Hospital, Ministry of Health, Makkah 24382, Saudi Arabia; 9Department of Pharmacy Administration and Clinical Pharmacy, Xi’an Jiaotong University, Xi’an 710061, China; 10Discipline of Clinical Pharmacy, School of Pharmaceutical Sciences, Universiti Sains Malaysia, Gelugor 11800, Malaysia; 11Department of Pharmacy Services, District Headquarter (DHQ) Hospital, Pakpattan 57400, Pakistan; 12Punjab University College of Pharmacy, University of the Punjab, Lahore 54000, Pakistan; 13Faculty of Pharmaceutical Sciences, UCSI University, Kuala Lumpur 56000, Malaysia; 14Children’s Hospital and University of Child Health Sciences, Lahore 54000, Pakistan; 15Department of Pharmacology and Toxicology, Faculty of Medicine in Al-Qunfudah, Umm Al-Qura University, Makkah 21955, Saudi Arabia; 16Department of Pharmacology and Toxicology, College of Pharmacy, Umm AL-Qura University, Makkah 21955, Saudi Arabia; 17Pharmaceutical Care Department, Alnoor Specialist Hospital, Ministry of Health, Makkah 24382, Saudi Arabia; 18Department of Clinical Pharmacy, College of Pharmacy, Prince Sattam bin Abdulaziz University, Alkharj 11942, Saudi Arabia

**Keywords:** antimicrobial resistance, antimicrobial stewardship programs, antimicrobial utilization, COVID-19, irrational use of antibiotics, Pakistan, point prevalence survey

## Abstract

The COVID-19 pandemic has significantly influenced antimicrobial use in hospitals, raising concerns regarding increased antimicrobial resistance (AMR) through their overuse. The objective of this study was to assess patterns of antimicrobial prescribing during the current COVID-19 pandemic among hospitals in Pakistan, including the prevalence of COVID-19. A point prevalence survey (PPS) was performed among 11 different hospitals from November 2020 to January 2021. The study included all hospitalized patients receiving an antibiotic on the day of the PPS. The Global-PPS web-based application was used for data entry and analysis. Out of 1024 hospitalized patients, 662 (64.64%) received antimicrobials. The top three most common indications for antimicrobial use were pneumonia (13.3%), central nervous system infections (10.4%) and gastrointestinal indications (10.4%). Ceftriaxone (26.6%), metronidazole (9.7%) and vancomycin (7.9%) were the top three most commonly prescribed antimicrobials among surveyed patients, with the majority of antibiotics administered empirically (97.9%). Most antimicrobials for surgical prophylaxis were given for more than one day, which is a concern. Overall, a high percentage of antimicrobial use, including broad-spectrums, was seen among the different hospitals in Pakistan during the current COVID-19 pandemic. Multifaceted interventions are needed to enhance rational antimicrobial prescribing including limiting their prescribing post-operatively for surgical prophylaxis.

## 1. Introduction

In the early 1990s, infectious diseases were the most common cause of death globally, which was addressed by the improved availability of effective antimicrobials [1,2,3]. Overall, antimicrobials have played a significant role in the treatment of infectious diseases since their development [4]. 

However, the irrational use of antibiotics increases antimicrobial resistance (AMR), increasing morbidity, mortality and associated costs as well as decreasing their effectiveness [5,6,7,8,9,10]. AMR is now seen as a major challenge globally that urgently needs addressing [11,12,13]. Consequently, it is very important that antimicrobials are prescribed and dispensed rationally across all sectors of care [14,15]. Evidence-based optimized antimicrobial therapy is necessary in order to ensure their effectiveness, safety and cost effectiveness as well as reducing the potential for AMR [8,16]. 

The first case of COVID-19 caused by severe acute respiratory syndrome coronavirus 2, (SARS-CoV-2) was identified in Wuhan, China [17]. The virus subsequently spread exponentially, affecting more than 626 million people world-wide with over 6.5 million deaths by the end of October 2022 [18]. The first case of COVID-19 was confirmed in Pakistan on February 26, 2020 [19], with cases appreciably increasing after that [20,21,22]. Various preventive measures were instigated across countries to try and limit the spread of the virus prior to the availability of effective treatments including vaccines. Public health measures included social distancing, closure of borders, mask wearing, avoiding crowded areas, hand hygiene, and quarantining of close contacts [23,24,25].

Recent publications have suggested substantial overuse of antimicrobials during the COVID-19 pandemic despite limited bacterial or fungal co-infections [26,27,28,29,30]. In addition, concerns that COVID-19 surges will increase the prevalence of hospital acquired infections, with their impact on increased mortality and costs [31,32,33]. In their study, Guan et al. (2020) ascertained that 58% of 1099 patients with COVID-19 received intravenous antibiotics [34]. A study in Brazil documented that intravenous antibiotics were administered in 84.7% hospitalized patients suffering from COVID-19 [35]. Recent reviews have suggested that between 61.8% and 74.6% of patients hospitalized with COVID-19 were prescribed antibiotics despite limited bacterial or fungal co-infections [27,28,36,37]; with prescribing rates varying across countries [37]. However, lower rates of antimicrobial prescribing have been seen in some countries [38]. These high rates of antimicrobial utilization may be due to prevalent symptoms (cough, fever), anxiety regarding the COVID-19 outbreak and concerns with secondary infections especially as this increases mortality [30,39]. However, high inappropriate use of antimicrobials in patients with COVID-19 enhances AMR; consequently, such prescribing should be avoided where possible [39,40,41,42,43,44,45,46].

Data on general hospital antimicrobial use, prescribing and resistance patterns during the COVID-19 pandemic is typically limited in low- and middle-income countries (LMICs) including Pakistan, although this is changing [22,45,47,48,49,50]. Antimicrobial stewardship programs (ASPs) are increasingly being introduced across countries to improve future prescribing [51,52,53]. This increasingly includes LMICs despite the many challenges they face, which include a lack of knowledge regarding ASPs as well as available resources [54,55,56,57,58]. This trend will continue.

Point prevalence surveys (PPS) provide a reliable method to ascertain the quantity and quality of antimicrobial prescribing in hospitals. Subsequently, the findings help to determine appropriate ASPs to instigate to improve future antimicrobial use [59,60,61,62]. Consequently, this survey was undertaken among different hospitals across Pakistan to evaluate the prevalence of patients receiving antimicrobial agents whilst in hospital during the current pandemic. This included the patterns of antimicrobial prescribing in all hospitalized patients and not just those with COVID-19. 

This is important since according to WHO guidelines, no antimicrobial or antifungal drug should be prescribed in patients with mild or moderate COVID-19 until confirmation of pre-existing symptoms of bacterial or fungal co-infections [63]. The Dutch Working Party also suggested sputum and blood cultures should be taken in patients with COVID-19 before starting antimicrobial therapy and recommended subsequent de-escalation if pertinent based on culture results [64]. However, we are aware that there can be challenges with undertaking routine culture and sensitivity testing among hospitals in LMICs due to issues of available facilities and costs including patient co-payments [65]. 

Our findings will build on recent PPS studies conducted in Pakistan including patients with COVID-19 [22,66,67,68,69,70,71]. The findings can subsequently be used to suggest potential initiatives, including ASPs, which can be instigated in Pakistan to improve future antimicrobial prescribing to reduce current high rates of AMR. This is critical given the concerns that currently exist surrounding the national action plan to reduce AMR in Pakistan [72]. In addition, currently extensive inappropriate antimicrobial use across all healthcare sectors in Pakistan, which is exacerbated by limited ASP activities to date [71,72,73,74,75]. 

## 2. Results

In total, 11 hospitals with 1810 beds were included in this PPS. In these hospitals, 1024 (56.57%) patients were in-patients on the day of the survey. Out of these, 662 (64.64%) patients were prescribed one or more antimicrobial. The overall number of antimicrobials prescribed were 1191. This works out at 1.76 antimicrobials per patent. The majority of patients prescribed antimicrobials were male (57.2%). Of all the antimicrobials prescribed, 500 (42%) were prescribed in the medical departments of hospitals, with the vast majority administered parenterally (92.2% of all administrations). 

For the management of infections, an appreciable number of antimicrobials were given for community acquired infections (71.9%). 21.2% of antimicrobials were prescribed for surgical prophylaxis, with most patients (64.4%) receiving antimicrobials for more than one day. Of concern is that only one culture report was observed at the time of the survey, with almost all antibiotics prescribed empirically (97.9%). In an appreciable number of cases, the reasons for prescribing antibiotics were not documented in the patients’ medical files (89.8% of patients’ files) (Table 1). 

The total number of prescribed antimicrobials for systematic use was 1163 (97.6%), of whom 518 (43.5%) were cephalosporins and carbapenems (J01D) (Table 2). The other antimicrobials prescribed include antivirals (Table 2).

The most common clinical indications where antimicrobials were prescribed were pneumonia (13.3%), central nervous system infections (10.4%), gastrointestinal prophylaxis (10.4%), and obstetric and gynecological prophylaxis (10.2%). The top four most commonly prescribed antimicrobials were ceftriaxone –‘Watch’ antibiotic (26.6%), metronidazole—‘Access’ antibiotic (9.7%), vancomycin —‘Watch’ antibiotic (7.9%) and meropenem–Watch’ antibiotic (5.5%) (Table 3).

## 3. Discussion

We believe this is the first study undertaken in Pakistan that documents the prevalence of antimicrobial use among hospitals, including those with or without suspected or actual COVID-19, during the current pandemic. In our study, 64.64% of admitted patients had been prescribed at least one antimicrobial. However, it is difficult to compare this with other LMICs since most recent published studies, including systematic reviews, have concentrated on the prescribing of antimicrobials in patients with actual or suspected COVID-19 rather than all admitted patients. Typically, appreciable antimicrobial prescribing has been seen in patients admitted with COVID-19 across countries, averaging between 61.8% and 74.6% of all patients hospitalized with COVID-19 in recent studies despite limited bacterial or fungal co-infections [27,28,30,36,37,76]. However, lower rates of antimicrobial prescribing in patients with COVID-19 have been seen in some high-income countries. This includes Scotland at 38.3% [77], and Singapore at only 6.2% [38]. 

The average number of antimicrobials prescribed per patient in our study was 1.76. This was similar to the findings in Bahawal Victoria Hospital, Pakistan, at 1.4 per patient [78], and 1.64 per patent in a previous PPS study involving 13 hospitals in Pakistan [69]. However, both studies were pre-pandemic. Similar to a previous PPS study in Pakistan pre-pandemic, common indications for prescribing were prophylaxis for obstetrics and gynecology procedures as well as for gastrointestinal conditions [69]. There was similarly an appreciable administration of antimicrobials parenterally. This could be explained by the fact that among hospitalized patients admitted for COVID-19, injectable antimicrobials were prescribed to prevent deterioration despite limited bacterial or fungal co-infections [66]. This mirrors the situation seen among patients admitted to hospitals in Pakistan with COVID-19 [66,67].

The majority of antimicrobials in our study were prescribed for the treatment of infections. This may be due to a long stay among patients admitted to the hospital with COVID-19, increasing the risk of acquiring infections [79,80,81]. However, we could not investigate this further in our current study. Further studies are needed to assist with this as part of future quality improvement programs. Of concern is that the vast majority of antimicrobial prescriptions in our study were empiric appreciably increasing the chances of inappropriate prescribing. This though was similar to other studies in Pakistan as well as other LMICs in patients with or without COVID-19 [28,36,69,82,83,84,85,86]. In addition, an appreciable number of medical records (89.8%) were without any documentation of the rationale for prescribing. Increasing the documentation for the indication, as well as inserting start, stop and review dates, is an important step for good antimicrobial prescribing practices to limit inappropriate prescribing and reduce AMR [87,88,89,90]. There is also concern regarding current adherence to prescribing guidance in our study, with adherence to guidance increasingly seen as good-quality prescribing [60,91,92,93]. Adherence to guidelines should be increased by the ongoing development, dissemination, and subsequent monitoring of adherence to new national guidelines currently being developed via the National Institute of Health in Pakistan (Appendix A).

To minimize the development of AMR, optimizing the course of treatment, including limiting the prescribing of antimicrobials post-operatively to reduce surgical site infections (SSIs), is important [16,94]. This includes promoting the prescribing of ‘Access’ antibiotics versus ‘Watch’ or ‘Reserve’ antibiotics from the ‘AWaRe’ list [95,96], with the over prescribing of ‘Watch’ antibiotics seen as problematic among LMICs [59]. In all hospitals, the cephalosporins, including ceftriaxone, were commonly prescribed antimicrobials because of their wide spectrum and safety. This is similar to previous studies conducted among LMICs [59,97,98,99]. A study in Bangladesh reported the same results on their COVID-19 wards [47]. In the present study, ceftriaxone was the most prescribed antimicrobial. This is a concern, with the prescribing of ceftriaxone generally increasing in Pakistan in recent years, similar to a study in the USA [100]. This is because such prescribing can increase extended spectrum beta lactamase (ESBL) producing multidrug resistant microbes [101]. The WHO have categorized ceftriaxone as a ‘Watch’ antibiotic in their AWaRe classification in view of the resistance potential [59,95,96], which should be reflected in any future national prescribing guidance produced in Pakistan. In addition, a key target for future ASPs in Pakistan should be to reduce inappropriate prescribing of ceftriaxone replaced by increased prescribing of pertinent ‘Access’ antibiotics. We have seen proactive ASPs and other approaches increasingly instigated among LMICs to improve future antimicrobial prescribing (Appendix A) across sectors, which includes reviewing the prescribing of colistin—a ‘Reserve’ antibiotic [16,52,74,94,102,103,104,105], These examples (Appendix A) provide future guidance to all key stakeholders in Pakistan.

We are aware of a number of limitations with this study. Firstly, we did not assess the appropriateness of antimicrobial use among all patients including those with COVID-19. This was hampered by the unavailability of local guidelines and lack of documentation in patients’ medical charts including a lack of CST findings. However, this is typical for PPS studies in LMICs. Similarly, we could only record information contained in the patients’ notes. We also did not differentiate between patients with or without COVID-19 in our study. This might have increased the prevalence rates of those without COVID-19. Finally, the number of hospitals actually taking part in the study were not fully representational of the current situation in Pakistan since a number of initially approached hospitals were not able to participate due to a variety of reasons. Nevertheless, we believe this study provides details of the patterns of antibiotic use among hospitals in Pakistan during COVID-19 and key areas to concentrate on for future ASPs. We will now be following this up. 

## 4. Materials and Methods

### 4.1. Study Design and Settings

A point prevalence survey of antimicrobial was undertaken across hospitals in Pakistan using the Global-PPS methodology from November 2020 to January 2021 [60,106]. The objective is to evaluate and document prevalence and prescribing patterns of antimicrobials among a number of hospitals during the current COVID-19 pandemic. This included patients with or without COVID-19 admitted to hospital. In total, 41 hospitals from secondary and tertiary care sectors, either private or public groups, were initially invited from the different cities throughout Pakistan to participate in this PPS using a purposeful sampling approach. We included both private and public hospitals to reflect the current situation in Pakistan, similar to PPS studies undertaken in other countries [107,108]. Participation of the hospitals was voluntary. Overall, twenty-one hospitals from different cities throughout Pakistan agreed to participate in this PPS study. However, bed occupancy of six secondary care hospitals was extremely low because of the current COVID-19 pandemic. Moreover, patients typically do not stay overnight in these settings. Consequently, these six hospitals were excluded from the final list of participating hospitals. Likewise, health care facilities having only nursing care, rehabilitation centers, or psychiatric centers were also excluded. Finally, 11 hospitals were included in this current PPS reflecting previous PPS studies in Pakistan [61,109].

### 4.2. Instrument of Measure

In order to collect data at the hospital, ward and patient level, uniform paper data collection forms were used from the Global PPS design [59,60]. The hospital information form in the Global PPS studies included the category of the participating hospital, the number of departments, the number of patients hospitalized and bed capacities. The medicine formulary of each hospital was checked before data collection with respect to antimicrobial availability in hospitals as there can be shortages, which is similar to other LMICs [110,111,112,113]. The ward data form consisted of total bed capacity, department specialty and the total number of patients that were admitted. The patient data form comprised their gender, prescribed antibiotics and their dosage regimen, the number of antimicrobials per patient, reasons for antibiotic prescribing and causative microorganisms if documented. A web-based program designed by University of Antwerp was used for data entry, validation and reporting [110].

With respect to guideline availability and adherence, antimicrobial guidelines have not been developed nationally within Pakistan or typically among the institutions surveyed. In their absence, institutions typically follow prescribing guidance for dosage and indications included in the British National Formulary [94,114]; alternatively, currently do not follow any guidelines. This is being addressed with national guidelines currently being developed through the National Institute of Health in Pakistan.

### 4.3. Inclusion and Exclusive Criteria

Patients who were receiving at least one antimicrobial (antibacterial, antifungal or antiviral) for their clinical condition for systemic use were included in the PPS. Short-stay patients, discharged patients and patients in emergency, outpatient departments and long-term care units were excluded in line with other PPS studies [69,109].

### 4.4. Data Collection

As mentioned, a structured data collection tool (Global PPS) was used to collect the data. Prescribing charts and patient’s medical case notes were checked for detailed information regarding the variables of interest. All patients hospitalized overnight and remained in the ward at 08:00 am on the day of the PPS were included. All the prescribed antimicrobials at the time of the survey were included. The data was double-checked for accuracy and completeness. All the definitions of medical treatment including surgical prophylaxis, healthcare associated infections (HAI) and community acquired infections were taken from Global PPS method [110]. There was no contact with any patient at any stage of data collection. The collected data were entered onto the web-based Global-PPS program. The Anatomical Therapeutic Chemical (ATC) classification system of the WHO was used to classify the different antimicrobials [115]. In addition, the principal antibiotics prescribed were broken down by their AWaRe classification, with the prescribing of antibiotics from the ‘Access’ list preferred to those from the ‘Watch’ and ‘Reserve’ list to reduce resistance potential [59,60,96,116]. 

### 4.5. Antimicrobial Stewardship Program Exemplars

A narrative review was undertaken by the co-authors to document exemplars of ASPs undertaken among LMICs and their outcome to provide future guidance. We have used this approach before when providing examples [16,94,105,117,118]. The LMICs will be broken down into their World Bank Classification, building on previous publications, since there have been concerns regarding the ability of LMICs to undertake ASPs due to resource issues [54,105,119]. However, this is now less of an issue [55,56].

### 4.6. Statistical Analysis

Data were analyzed using the Microsoft Excel and Statistical Package for the Social Sciences (SPSS) and descriptive statistics were applied.

## 5. Conclusions

There were concerns with high levels of prescribing of antibiotics among hospitals in Pakistan during the COVID-19 pandemic. Empiric prescribing dominated, with ceftriaxone the most commonly prescribed antibiotic. This is a concern as such prescribing may well enhance already high rates of AMR in Pakistan. Various strategies and initiatives are required to improve future prescribing of antibiotics among hospitals in Pakistan. This includes the routine introduction of ASPs among hospitals throughout Pakistan as well as increasing the capacity of hospitals to routinely undertake culture and sensitivity testing. ASPs can incorporate educational programs surrounding the rational prescribing of antibiotics, continuous supervision and feedback. Key areas for ASPs include increasing the documentation of the indication for prescribing, reducing empiric prescribing, reducing the extent of antibiotics prescribed post-operatively to prevent SSIs and increasing the prescribing of ‘Access’ antibiotics where pertinent as part of newly developed national guidelines. In addition, increasing the prescribing of oral antibiotics via de-escalation where applicable to reduce hospital stay and costs.

## Figures and Tables

**Table 1 antibiotics-12-00070-t001:** Overall antibiotic use prevalence.

Characteristics N (%)	Total
Total beds	1810
Hospitalized patients	1024
Number of treated patients	662 (64.64)
Number of prescribed antibiotics	1191 (1.76/patient)
**Departments**	
Surgical ward	286 (24)
Medical ward	500 (42)
Intensive care unit	67 (5.6)
Pediatric medical ward	331 (27.8)
Pediatric intensive care unit	7 (0.6)
**Gender**	
Male	681 (57.2)
Female	510 (42.8)
**Route of administration (where recorded)**	
Oral	89 (7.5)
Parenteral	1098 (92.2)
**Indication**	
Community-acquired infection	856 (71.9)
Hospital-acquired infection	20 (1.7)
Medical prophylaxis	30 (2.5)
Surgical prophylaxis (single dose)	5 (0.4)
Surgical prophylaxis (one day)	85 (7.1)
Surgical prophylaxis (>1 day)	163 (13.7)
Others	17 (1.4)
**Treatment**	
Empirical therapy	1166 (97.9)
Targeted therapy	25 (2.1)
**Guideline compliance**	
Yes	12 (1.0)
No	662 (55.6)
NA	466 (39.1)
NI	51 (4.3)
**Stop date documented**	118(9.9)
Reason on notes Yes No	122 (10.2) 1069 (89.8)
**Culture Reports**	1

Empirical therapy: Treatment given without finding out the causative microbe; Targeted therapy: Treatment given after finding the causative microbe; NA: Not assessable because of the absence of any guidelines; NI: No information because of incomplete patient history.

**Table 2 antibiotics-12-00070-t002:** Prevalence of the main antimicrobial classes.

Antibiotics	N (%)
**ANTIBACTERIALS FOR SYSTEMIC USE** (J01)	1163 (97.6)
Tetracyclines (J01A)	4 (0.3)
Amphenicols (J01B)	0 (0)
Penicillins (J01C)	147 (12.3)
Cephalosporins and carbapenems (J01D)	518 (43.5)
Sulfonamides and trimethoprim (J01E)	2 (0.2)
Macrolides and lincosamides (J01F)	54 (4.5)
Aminoglycosides (J01G)	129 (10.8)
Quinolones (J01M)	59 (5.0)
Other antibacterial (J01X)	247 (20.7)
ANTIMYCOTICS FOR SYSTEMIC USE (J02)	0 (0)
ANTIMYCOBACTERIALS FOR SYSTEMIC USE (J04)	11 (0.9)
ANTIVIRALS FOR SYSTEMIC USE (J05)	13 (1.1)
ANTIPROTOZOALS (P01)	0 (0)

**Table 3 antibiotics-12-00070-t003:** Top 10 indications and antibiotics.

	Top 10 Indications		Top 10 Antibiotics	
	Indications	N (%)	Antibiotics	N (%)
**1.**	Pneumonia	158 (13.3)	Ceftriaxone	317 (26.6)
**2.**	CNS	124(10.4)	Metronidazole	115 (9.7)
**3.**	GIT-P	124(10.4)	Vancomycin	94 (7.9)
**4.**	OBGY	121 (10.2)	Meropenem	66 (5.5)
**5.**	BJ	93 (7.8)	Ciprofloxacin	34 (2.9)
**6.**	CVS	78 (6.5)	Piperacillin, enzyme inhibitor	14(1.2)
**7.**	COVID-19	59 (5.0)	Levofloxacin	10(0.8)
**8.**	SST	50 (4.2)	Cefipime	6 (0.5)
**9.**	SEPSIS	27 (2.6)	Cefuroxime	4(0.3)
**10.**	Unknown	7 (0.6)	Clindamycin	2 (0.2)

BJ, bone and joint; CNS, central nervous system; GIT, gastro-intestinal tract; OBGY, obstetrics and gynecology; P, prophylaxis; SST, skin and soft tissues.

## Data Availability

Further data are available upon reasonable request from the corresponding authors.

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
