# Peer review of "Point Prevalence Survey of Antimicrobial Use during the COVID-19 Pandemic among Different Hospitals in Pakistan: Findings and Implications"

_antibiotics, 2022, doi:10.3390/antibiotics12010070_

Round 1

Reviewer 1 Report

Line 60 Remove comment: “We will now be following this up”

Date of the PPS

Time of the PPS

Explain the COVID-19 situation in Pakistan:

Number of cases

Variants

Waves

The number of COVID-19 cases included in the PPS.

Comments on country microbiology/resistance patterns.

Comments on local (hospital) microbiology/resistance patterns.

Comments on bacteriology/resistance of bacteria isolated from healthcare-associated infections locally and nationally.

Comments on resistance patterns before, during, and on the day of the  PPS.

Lines 183-185 are confusing.

Reference 61 is a duplicate of reference 71.

Add and review the following references:

Baker MA. Clin Infect Dis 2022; 74:1748.

Langford BJ. Clin Microbiol Infect 2021; 27:520.

Lai CC. Int J Antimicrob Agents 2021; 57:106324.

Choudhury S. Bull World Health Organ 2022; 100:295a.

Baggs J. Clin Infect Dis 2022; 75:s294.

Baghdadi JD. Antimicrob Agents Chemother 2021; 65:e134121.

Mendez Neto AG. J Med Virol 2021; 93:1489.

Author Response

English language and style

( ) English very difficult to understand/incomprehensible
(x) Extensive editing of English language and style required
( ) Moderate English changes required
( ) English language and style are fine/minor spell check required
( ) I don't feel qualified to judge about the English language and style

Author comment: Thank you for this. The paper has now been updated with the help of one of the co-authors who is a native English speaker with over 450 papers in peer-reviewed Journals since 2008. We hope this is now acceptable.

Comments and Suggestions for Authors

1) Line 60 Remove comment: “We will now be following this up”

Author comment: Thank you – now removed.

2) Date of the PPS

Author comment: Thank you for your comment. The date and time of PPS study has now been added to the Materials and Methods section.

3) Time of the PPS

Author comment: Thank you for your comment. The date and time of PPS study has now been added to the Materials and Methods section.

4) Explain the COVID-19 situation in Pakistan:

Number of cases; Variants; Waves; The number of COVID-19 cases included in the PPS.

Author comment: Thank you for your comment. The number of cases, variants, and waves are the dynamic figures which were changing abruptly during the course of our study. The main focus of this PPS was to highlight the use of antimicrobials during the current COVID-19 pandemic, not to discuss the situation and clinical presentation of COVID-19 vs. non-COVID-19 patients in Pakistan. Just concentrating on patients with COVID-19 would be out of the scope of this current study, and we hope this is acceptable

5) Comments on country microbiology/resistance patterns.

Author comment: Thank you for your comment. Based on limited published data, there is high level of resistance against the majority of commonly used antimicrobials in Pakistan. In Pakistan, 3rd generation cephalosporins are the most commonly prescribed antibiotics with no proper prophylactic guideline. This needs to be addressed because this irrational use is burdening the healthcare system and driving up resistance rates. We have made suitable comments regarding ASPs to emphasize this point, and hope this is now acceptable.

6) Comments on local (hospital) microbiology/resistance patterns.

Author comment: Thank you for your comment. No prominent data is available on the individual local (hospital) microbiology/resistance patterns due to overburdening of hospitals with patients and lack of resources to provide assistance in conducting microbial cultural tests on a large scale. This is seen in a number of LMICs. However – this needs to be addressed going forward, and we have now included this in our conclusions. We hope this is acceptable,

7) Comments on bacteriology/resistance of bacteria isolated from healthcare-associated infections locally and nationally.

Author comment: Thank you for your comment. No prominent data regarding bacteriology/resistance of bacteria isolated from healthcare-associated infections locally and nationally was available during this study due to overburdening of hospitals with patients and lack of resources to provide assistance in conducting microbial cultural tests on a large scale. This is in line with other LMICs with high rates of empiric prescribing (now referenced). We hope this is acceptable.

8) Comments on resistance patterns before, during, and on the day of the  PPS.

Author comment: Thank you for your comment. No data is available on this and this wasn’t the objective of PPS study. This is similar to other PPS studies where the emphasis is on ascertaining current utilization patterns as well as adherence to agreed quality indicators. In addition, we can only document the data actually recorded in the notes. This typically did not include data on resistance patterns. Consequently, we cannot comment on this. We hope this is OK.

9) Lines 183-185 are confusing.

Author comment: Thank you – now amended, We hope this is now acceptable.

10) Reference 61 is a duplicate of reference 71 and add and review the following references:

Baker MA. Clin Infect Dis 2022; 74:1748; Langford BJ. Clin Microbiol Infect 2021; 27:520; Lai CC. Int J Antimicrob Agents 2021; 57:106324; Choudhury S. Bull World Health Organ 2022; 100:295a; Baggs J. Clin Infect Dis 2022; 75:s294; Baghdadi JD. Antimicrob Agents Chemother 2021; 65:e134121; Mendez Neto AG. J Med Virol 2021; 93:1489.

Author comment: Thank you for your comment. In fact, old References 61 and 71 are different – with one referring to cancer patients and the other to all patients. We have now included these extra references in the revised text (in fact 2 had already been included – Langford et al and Lai et al), and hope this is now acceptable

Reviewer 2 Report

Dear Authors,

The article titled “Point prevalence survey of antimicrobial use during the COVID-19 pandemic among different hospitals in Pakistan: Findings and Implications”, presents an important local review to quantify the impact of antibiotic use as a result of the pandemic. Although the document has a moderate impact on readers, due to the multitude of current studies with a similar approach, this reviewer considers that after an exhaustive revision of the English it is possible to be published in the journal. It is suggested to submit the document to an English certification before being published.

Author Response

English language and style

( ) English very difficult to understand/incomprehensible
(x) Extensive editing of English language and style required
( ) Moderate English changes required
( ) English language and style are fine/minor spell check required
( ) I don't feel qualified to judge about the English language and style

Author comment: Thank you for this. The paper has now been updated with the help of one of the co-authors who is a native English speaker with over 450 papers in peer-reviewed Journals since 2008. We hope this is now acceptable.

Comments and Suggestions for Authors

Dear Authors,

The article titled “Point prevalence survey of antimicrobial use during the COVID-19 pandemic among different hospitals in Pakistan: Findings and Implications”, presents an important local review to quantify the impact of antibiotic use as a result of the pandemic.

Author comments: Thank you for these encouraging words – appreciated

Although the document has a moderate impact on readers, due to the multitude of current studies with a similar approach, this reviewer considers that after an exhaustive revision of the English it is possible to be published in the journal. It is suggested to submit the document to an English certification before being published.

Author comments: Thank you for this. The paper has now been revised by one of the co-authors who is a native English speak with over 450 publications in peer-reviewed Journals since 2008. We hope this is now OK.

Reviewer 3 Report

Dear Authors, thank you for giving me the opportunity to review your manuscript. The general impression made by this publication - it is “raw”, immature and should be revised substantially. Please find some comments below:

·       The paper has totally no link between COVID-19 and antibiotic use. There is no description on the percentages of patients with actual/suspected COVID-19. It lacks profound data and results. Discussion and conclusions parts are scarce.

·       The paper doesn’t show any actual dynamics regarding the declared in introduction «significantly influenced antimicrobial use». Neither it doesn’t assess patterns of antimicrobial prescribing as totally no information may be utilized from the Tables (e.g Table 3). Should there be an assumption that metronidazole was the second used antibiotic for the patients with pneumonia (top 1 indication) in Pakistan? What kind of pneumonia (bacterial/viral) was it anyway?

·       Please provide the characteristics of the hospitals (multi-field/specialized etc.) included in the Point prevalence survey (PPS). Were there any COVID-19 wards on the day of the Study? Why were the hospitals only half-full (56,6%) during the COVID-19 pandemic?

·       Please specify the exact time period when the PPS was conducted.

·       Table 1: Please specify what is the medical department, medical prophylaxis. In treatment, please distinguish antibiotics and antivirals, empiric and targeted treatment.

·       Table 2: Please specify 20% of other antibacterials once you show 2 cases of sulfonamides&trimethoprim use (0,2%).

·       There were no Table S1 in the text/provided materials.

·       Please provide the paper data collection forms that were used to collect the data in the Supplementary.

·       The language of the publication along with the spelling should be revised and corrected substantially.

Author Response

English language and style

( ) English very difficult to understand/incomprehensible
(x) Extensive editing of English language and style required
( ) Moderate English changes required
( ) English language and style are fine/minor spell check required
( ) I don’t feel qualified to judge about the English language and style

 Author comments: Thank you for this. The paper has now been revised by one of the co-authors who is a native English speak with over 450 publications in peer-reviewed Journals since 2008. We hope this is now OK.

Comments and Suggestions for Authors

Dear Authors, thank you for giving me the opportunity to review your manuscript. The general impression made by this publication - it is “raw”, immature and should be revised substantially. Please find some comments below.

 Author comments: Thank you for your comments. Hopefully, we have adequately updated the paper

1) The paper has totally no link between COVID-19 and antibiotic use. There is no description on the percentages of patients with actual/suspected COVID-19. It lacks profound data and results. Discussion and conclusions parts are scarce.

 Author comments: Thank you for your comment. The focus of this PPS was to highlight the use of antimicrobials during the COVID-19 not to discuss the situation regarding the management of patients with COVID-19 in Pakistan (which we have published on separately). Discussing this would be out of the scope of our study. We have made this point more strongly, and hope this is now acceptable.

2) The paper doesn’t show any actual dynamics regarding the declared in introduction «significantly influenced antimicrobial use». Neither it doesn’t assess patterns of antimicrobial prescribing as totally no information may be utilized from the Tables (e.g Table 3). Should there be an assumption that metronidazole was the second used antibiotic for the patients with pneumonia (top 1 indication) in Pakistan? What kind of pneumonia (bacterial/viral) was it anyway?

 Author comments: Thank you for your comment. No such information was present on the patient’s clinical notes about the differential diagnosis of the pneumonia (this is typical in PPS studies in LMICs where often key information is lacking – with PPS studies providing a basis for instigating future quality improvement programmes). The data was specifically collected from the patient’s file and it was not possible to asses the type of pneumonia patient was suffering from. We hope this is OK.

Table 3 describes two different scenarios. One is the top 10 indications and other is the top 10 antibiotics. The reader is not supposed to link these two different scenarios with each other, and we have reported on them separately especially regarding potential ways to reduce unnecessary prescribing of ‘Watch’ antibiotics to reduce high and growing rates of AMR in Pakistan. We hope this is also now OK.

3) Please provide the characteristics of the hospitals (multi-field/specialized etc.) included in the Point prevalence survey (PPS). Were there any COVID-19 wards on the day of the Study? Why were the hospitals only half-full (56,6%) during the COVID-19 pandemic?

 Author comments: Thank you for your comment. All the hospitals were general (multifield). There was no data available on the COVID-19 wards on the day of the study. The reason hospitals were only half full during the COVID-19 pandemic was the general fear among population about being quarantined on unknown places (wards) if they were positive. So, people were hesitating visiting the hospitals during that period (which we comment on in the Methods section resulting in less hospitals being included in this PPS study than initially envisaged).

4) Please specify the exact time period when the PPS was conducted.

 Author comments: Thank you for your comment. The date and time of PPS study has now been added to the Materials and Methods section.

5) Table 1: Please specify what is the medical department, medical prophylaxis. In treatment, please distinguish antibiotics and antivirals, empiric and targeted treatment.

 Author comments: Thank you for your comment, the details have been added below Table 1, and we hope this is now OK.

6) Table 2: Please specify 20% of other antibacterials once you show 2 cases of sulfonamides & trimethoprim use (0,2%).

 Author comments: Thank you for your comment. Other antibacterials (J01X) is the name of the classification named by the WHO. There are numerous other antibacterials present in them. It is a complete class just like Tetracyclines. We hope this is OK

7) There were no Table S1 in the text/provided materials.

Author comments: Thank you – this was included just before the references. In the final paper (if approved) there will be a URL link.

8) Please provide the paper data collection forms that were used to collect the data in the Supplementary.

 Author comments: Thank you – these are the standard extensive forms that can be found on the internet (together with the manual) for Global PPS studies (https://www.global-pps.com/documents/). We hope this is now OK as it would be impractical to include these in the Supplementary Material.

9) The language of the publication along with the spelling should be revised and corrected substantially.

 Author comments: Thank you for this. The paper has now been revised by one of the co-authors who is a native English speak with over 450 publications in peer-reviewed Journals since 2008. We hope this is now OK.

Round 2

Reviewer 1 Report

Good work

Reviewer 3 Report

Dear Authors! Thank you for your revised version of the manuscript. Although the language has been substantially revised the general significance of content remained low. As I've mentioned before, the paper has no link with COVID-19 as may be assumed by the title. It doesn't show any dynamics of pre-pandemic and COVID-19 pandemic use of antibiotics. Neither it demonstrates any particular/specific gaps to reduce unnecessary prescribing or to reduce growing rates of antimicrobial resistance in Pakistan. The design of the study gives rather a general idea of total antibiotic consumption than any indication-oriented prescribing in Pakistan to provide  possible interventions to improve antimicrobial stewardship. Unfortunately the study lacks scientific soundness limiting the conclusions that can be drawn.